# The Effect of Antibiotics on the Infant Gut Fungal Microbiota

**DOI:** 10.3390/jof8040328

**Published:** 2022-03-22

**Authors:** Rebecka Ventin-Holmberg, Schahzad Saqib, Katri Korpela, Anne Nikkonen, Ville Peltola, Anne Salonen, Willem M. de Vos, Kaija-Leena Kolho

**Affiliations:** 1Human Microbiome Research Program, Faculty of Medicine, University of Helsinki, 00014 Helsinki, Finland; rebecka.ventin-holmberg@helsinki.fi (R.V.-H.); schahzad.saqib@helsinki.fi (S.S.); katri.korpela@helsinki.fi (K.K.); anne.salonen@helsinki.fi (A.S.); willem.devos@wur.nl (W.M.d.V.); 2Folkhälsan Research Center, 00250 Helsinki, Finland; 3Children’s Hospital, Helsinki University, 00029 Helsinki, Finland; anne.nikkonen@hus.fi; 4Department of Paediatrics and Adolescent Medicine, Turku University Hospital, University of Turku, 20014 Turku, Finland; vilpel@utu.fi; 5Laboratory of Microbiology, Wageningen University, 6708 WE Wageningen, The Netherlands; 6Department of Pediatrics, Tampere University, 33520 Tampere, Finland

**Keywords:** amoxicillin, bacteria, *Candida*, children, macrolide, mycobiota

## Abstract

Antibiotics are commonly used drugs in infants, causing disruptions in the developing gut microbiota with possible detrimental long-term effects such as chronic inflammatory diseases. The focus has been on bacteria, but research shows that fungi might have an important role as well. There are only a few studies on the infant gut fungal microbiota, the mycobiota, in relation to antibiotic treatment. Here, the aim was to investigate the impact of antibiotics on the infant gut mycobiota, and the interkingdom associations between bacteria and fungi. We had 37 antibiotic-naïve patients suffering from respiratory syncytial virus, of which 21 received one to four courses of antibiotics due to complications, and 16 remained antibiotic-naïve throughout the study. Fecal samples were collected before, during and after antibiotic treatment with a follow-up period of up to 9.5 months. The gut mycobiota was studied by Illumina MiSeq sequencing of the ITS1 region. We found that antibiotic use affected the gut mycobiota, most prominently seen as a higher relative abundance of *Candida* (*p* < 0.001), and a higher fungal diversity (*p* = 0.005–0.04) and richness (*p* = 0.03) in the antibiotic-treated infants compared to the antibiotic-naïve ones at multiple timepoints. This indicates that the gut mycobiota could contribute to the long-term consequences of antibiotic treatments.

## 1. Introduction

The gut microbiota is known to include bacteria, fungi, archaea, viruses and sometimes protozoa and is a major factor contributing to human health [1]. Most microbiota-related studies have focused on the bacteria, although the fungal microbiota, the mycobiota, might have a significant role as well. Imbalance in the intestinal fungal composition has been associated with diseases such as inflammatory bowel disease (IBD) [2,3,4,5,6], celiac disease [7] and colorectal cancer [8]. The antifungal drug fluconazole worsens inflammation in mice with induced intestinal- and lung inflammation, indicating that the fungal microbiota also has a protective role [9].

Compared to the bacterial gut microbiota, the gut mycobiota has fewer taxa and seems to be less stable [10,11]. Variation both between and within individuals makes it difficult to determine whether certain fungi are actual colonizers or only transient fungi derived from food or environment as previously reviewed [12]. The gut mycobiota consists of the phyla Ascomycota and Basidiomycota, with Ascomycota being more abundant in both adults [11,13,14,15] and in children [16].

The first major exposure to microorganisms is through birth, and the infant gut has been observed to be colonized by fungi [17] as early as 10 days after birth [16]. The gut microbiota continues to develop during the first years of life, affected by multiple factors such as delivery mode and diet [14,18], and particularly the use of antibiotics [19]. At two years of age, the infant gut mycobiota shows similarities to that of adults [16].

Many newborns receive antibiotics due to an infection during early years of life. In Finland, 26.5% of children < 5 years of age received at least one course of antibiotics in 2019 (Kela statistics, www.kela.fi/tilastot, accessed on 2 February 2022; and statistics in Finland, www.stat.fi/til, accessed on 2 February 2022). The use of antibiotics during the critical time from conception to 2 years of age has been reported to cause both short-term problems including dramatic shifts of the gut fungal and bacterial microbiota and even sepsis, and further long-term consequences such as asthma, atopy, pediatric onset of Crohn’s disease and metabolic diseases [20,21]. The use of antibiotics is also associated with higher body mass index (BMI) [19,22,23,24], and in a study conducted on Finnish pre-school children the use of antibiotics was associated with increased risk for overweight and asthma later in life [25,26].

Antibiotics, evidently only affecting prokaryotes, may induce fungal growth, particularly of *Candida* spp. in the mouse gut [27,28,29], and cause major changes in the adult gut bacterial microbiota composition, with an even more persistent shift in the gut mycobiota [30]. Additionally, antibiotics have been observed to cause increased risk of candidiasis and bloodstream infection [31,32]. Bacteria and fungi co-exist in the gut, and competition for nutrient sources and the production of antifungal compounds by gut bacteria and antibacterial compounds by fungi are characteristic of the interkingdom interactions [30,33,34]. *Candida* is a part of the healthy gut mycobiota [13], but simultaneously it seems to be a driver of aberrant composition of gut microbiota and is associated with multiple diseases [2,3,4,5,6,7,8]. In a recent study it was observed that the decrease of short-chain fatty acids and anaerobic bacteria correlates with an increase of fungal growth [35].

There are many detrimental long-term effects in using antibiotics, particularly in children, that stem from an aberrant gut microbiota composition. Studies on the effect of antibiotics on the gut mycobiota are very limited. The impact of antibiotics on the gut bacterial microbiota has been published for these data [19] with the observation that the administration of only a single course of amoxicillin or macrolide already caused a consistent shift in the bacterial classes Actinobacteria, Gammaproteobacteria and Clostridia.

Here, our aim was to investigate the impact of antibiotics on the gut mycobiota of infants by investigating the gut mycobiota composition in antibiotic-treated compared to antibiotic-naïve infants. Further, we aimed to study the interkingdom correlations and to combine both fungal and bacterial gut microbiota.

## 2. Materials and Methods

### 2.1. Study Design

The cohort consisted of 37 children at a median age of 2 months who were hospitalized at the Children’s Hospital, University of Helsinki, or Turku University Hospital between December 2013 and May 2014 due to a respiratory syncytial virus (RSV) infection. Only antibiotic-naïve children were included. Fecal samples were collected before, during and after antibiotic treatment with a follow-up of up to 9.5 months. All fecal samples were immediately frozen at −20 °C until transported frozen to −70 °C where they were stored until analysis. At each sampling timepoint, questionnaires were filled out containing information on diet, probiotics and use of antibiotics. Additionally, information of birth mode, other possible diseases, growth and weight gain were available from the patient files. Preterm infants (birth < 37 gestational week or birth weight < 2500 g) or those with congenital malformations or syndromes were excluded. None of the infants had been traveling abroad. In analyses, the timepoints were calculated as time since the start of antibiotics, or time since the first sample for antibiotic-naïve patients. In the cases where multiple courses of antibiotics were administered to a patient, the days from the most recent antibiotic course were used in the timepoints. Therefore, the time of follow-up might be longer than the timepoint used in analyses.

### 2.2. DNA Extraction and MiSeq Library Preparation

The fecal samples were thawed, and the bacterial and fungal DNA was extracted using the bead-beating method as previously described [19]. The 16S rRNA gene amplicon data were obtained from the previous study by Korpela et al. [19]. The gut fungal microbiota composition was studied by Illumina MiSeq sequencing of the internal transcribed sequence 1 (ITS1) targeting a conserved region of fungi. The PCR primer pair ITS1F (FWD, CTTGGTCATTTAGAGGAAGTAA) and ITS2 (REV, GCTGCGTTCTTCATCGATGC) [36] were used to amplify the fungal DNA. The sequencing library was prepared as previously described [37]. Illumina MiSeq paired-end sequencing was performed in the Functional Genomics Unit, University of Helsinki, Helsinki, Finland. The ITS rDNA amplicon sequences of this study are available at the ENA database [PRJEB50378].

### 2.3. Analysis of Sequencing Data

The ITS MiSeq sequencing read data were processed according to the DADA2 ITS pipeline [38]. The reads were annotated as previously described [37] by annotating the amplicon sequence variants (ASVs) with BLAST [39]. The BLAST hits were limited to >75% query coverage and percentage identity and anything below this was discarded. The annotations were gut- or environment-specific. Before pre-processing, the median number of reads per sample was 66,791 (17,942–154,451). After pre-processing, the median read count was 7556 (134–32,846) and finally, after annotation the median read count was 1786 (121–22,932). Successful annotation to fungal taxa was obtained from 123 samples (out of 151 samples) from 37 patients and a total of 113 samples (75%) from 37 patients were annotated successfully with more than 100 reads.

### 2.4. Statistical Analysis

The R package mare [40] was used for analysis and visualizing the ITS relative abundance data, including the use of the packages vegan [41], MASS [42] and nlme [43]. *p*-values for taxon-specific differences were corrected for false discovery rate (FDR; Benjamini–Hochberg [44]). Generalized linear models with negative binomial distribution (glm.nb) from the MASS package [42] and Generalized Least Squares (gls) from the nlme package [43] were used to analyze differences in the microbiota composition between the control group and the antibiotic-treated patients. Samples with less than 100 reads were discarded from the analysis and sex, breastfeeding and hospital location were used as confounding variables within the models. Background variables were analyzed by multivariate permutational analysis of variance at both baseline and all timepoints, and the factors included in the statistical analyses of individual taxa were chosen based on their impact on the total fungal microbiota. The diversity was calculated as the inverse Simpson diversity index and richness as the number of ASVs. The Spearman correlations and *p*-values between fecal fungal and bacterial taxa at different timepoints were calculated with “rcorr” from the Hmisc package [45].

### 2.5. Ethics Statement

The study was approved by the Ethics committee of Helsinki and Uusimaa Hospital district. All guardians signed an informed consent to include their child in the study.

## 3. Results

### 3.1. Patient Characteristics

The aim was to study the effect of antibiotics on the gut mycobiota. We analyzed samples from 37 antibiotic-naïve infants who were recruited prospectively and included in our previous report on the bacterial gut microbiota [19]. All children were recruited while treated because of RSV infection. Out of these, 21 patients received antibiotics prescribed by a clinician due to complications such as otitis media. Sixteen infants did not receive any antibiotics and were considered controls in this study. The antibiotics used were amoxicillin and macrolides and the infants received one to four courses of antibiotics. Amoxicillin was given to 21 infants as their first antibiotic, and 8 of these received a second course of penicillin-group antibiotics and 4 patients received macrolides. The number of samples at each timepoint stratified by antibiotics used are presented in Table 1. The timepoints are calculated from the most recent course of antibiotics, or from the first sample for the controls.

Fungal amplicons generated from a total of 123 fecal samples obtained from the 37 recruited patients were sequenced. Out of these, 68 samples were derived from the 21 antibiotic-treated patients and 56 samples from the 16 control patients. Baseline samples of the antibiotic-treated patients were obtained from 12 patients. Background data of the study participants are presented in Table 2. During the follow-up (up to 9.5 months), there were no infants who developed any diseases.

### 3.2. Overview of the Gut Mycobiota Composition

Across all samples the fungal genus *Saccharomyces* was most prevalent (97%) with a relative abundance of 71%, followed by *Malassezia* (prevalence 44%) with a relative abundance of 8%, *Candida* (prevalence 30%) with a relative abundance of 9% and *Cladosporium* (prevalence 11%) with a relative abundance of 1%. The prevalence and relative abundance of all genera with a prevalence >5% is presented in Appendix A. The gut mycobiota composition at genus level was analyzed stratified by timepoints and patient groups (Figure 1). Location of hospital, sex and breastfeeding were used as background factors and the analyses were adjusted for these factors. All three background factors impacted the gut mycobiota significantly (*p* < 0.05) both at baseline and at all timepoints, except for the location of hospital, which was significant only when including all timepoints (Appendix A).

### 3.3. Difference between the Control Group and the Group Treated with Antibiotics

Before the start of antibiotic treatment there was no significant difference in gut mycobiota composition between the groups. At 1–2 days after the start of treatment (during treatment), the amoxicillin-treated group had significantly higher relative abundance of the fungal genus *Candida* (*p* FDR < 0.001, fold change = 132) (Figure 2A). When comparing antibiotic-treated patients (both amoxicillin and macrolides combined) 1–2 weeks after start of treatment to controls at 3–5 days after first sample, *Saccharomyces* was more abundant in the control group (*p* FDR < 0.001, fold change = 3.8) (Figure 2A). Further, *Saccharomyces* was more abundant in the control group at 3–5 days after the start of treatment compared to the antibiotic-treated patients (*p* FDR < 0.001, fold change = 1.44). At the end of the study (>6 weeks after start of treatment) the phylum Basidiomycota was more abundant in the antibiotic-treated group (both amoxicillin and macrolide combined) (*p* FDR = 0.009, fold change = 10.9) (Figure 2B). Detailed results of the statistical differences between the antibiotic-treated and antibiotic-naïve infants are presented in Appendix A.

When comparing the groups of antibiotics used, the amoxicillin-treated infants had significantly higher relative abundance of *Candida* at >6 weeks after start of treatment compared to controls (*p* FDR = 0.028, fold change = 13.14). This difference was not significant for macrolide-treated infants, although the trend was similar with non-FDR-corrected *p*-value < 0.05, and fold change 22. Both the amoxicillin- (*p* FDR < 0.001, fold change = 1.68), but notably the macrolide-treated (*p* FDR < 0.001, fold change = 48.45) infants had significantly higher relative abundance of *Malassezia* at 3–6 weeks after the start of antibiotic treatment. Further, Basidiomycota was more abundant in the macrolide-treated group compared to controls at 3–6 weeks after the start of treatment (*p* FDR = 0.033, fold change = 20.01).

Finally, the antibiotic-treated group had significantly higher fungal diversity and richness at 3–5 days (diversity, *p* = 0.04, richness, *p* = 0.03), 1–2 weeks ((when comparing to control at 3–5 days) diversity, *p* = 0.005, richness, *p* = 0.03)) and >6 weeks (diversity, *p* = 0.03, richness = 0.03) after the start of antibiotic treatment as seen in Figure 2C. At >6 weeks the difference in diversity was even more significant between the control and macrolide-treated group (*p* = 0.0027).

### 3.4. Spearman Correlations between Fungi and Bacteria

Interkingdom Spearman correlations were calculated at each timepoint for both the antibiotic-treated group and for those who remained antibiotic-naïve throughout the study. There were significant correlations (*p* < 0.05) at all timepoints, and these are presented in Appendix A. No robust interkingdom correlations were detected.

## 4. Discussion

Here we investigated the effect of antibiotics on the infant gut mycobiota. This was carried out by MiSeq sequencing of the PCR-amplified ITS1 region from the DNA of fecal samples collected from both controls and antibiotic-treated infants at six different timepoints with a follow-up ranging over 9 months. All patients were naïve to any antibiotics at the first timepoint and controls were antibiotic-naïve throughout the follow-up. We observed that the use of antibiotics affected the gut mycobiota, characterized by an increased relative abundance of *Candida* and higher diversity and richness in antibiotic-treated infants.

Antibiotics may have a detrimental effect on both the bacterial and fungal gut microbiota, with consequences such as antibiotic-associated diarrhea and recurrent *Clostridioides difficile* infection. In children, however, *C. difficile* infections after antibiotic use is rare [46]. The frequent use of antibiotics in infants and the many potential long-term effects, as presented in the introduction, however, highlight the importance to study the effects of antibiotics on the infant gut microbiota.

The healthy gut bacterial microbiota of infants is characterized by facultative anaerobic bacteria such as streptococci, enterococci and enterobacteria dominating the composition [47]. There are few studies on the infant gut mycobiota, and although the composition varies between studies, *Saccharomyces*, *Malassezia*, *Candida* and *Debaromyces* were reported to make up the core gut mycobiota of infants across these studies [10,16,17,48,49,50,51]. This is in line with our results, apart from *Debaromyces*, which interestingly was not one of the most abundant genera. In addition to the difficulty in distinguishing colonizers from transient fungi [12] of the gut, the methodologies used differ and this should also be considered when interpreting and comparing results [14]. In a Norwegian study following 298 healthy mother–offspring pairs, 56–76% of the infants had detectable fungi in their stool [16]. In our study we observed fungi in 75% of the samples. Since we have longitudinal sampling, we can observe that although we found high variability in the gut fungal composition both within patients and between patients, which is in line with a recent study [10], the most abundant fungi are present throughout the timepoints. This increases the probability that these are real colonizers of the infant gut. It is also worth noting that the infants in this study were not healthy, since both antibiotic-treated patients and controls had an active RSV infection at baseline and can therefore not fully be compared to the healthy infants in other studies. Viral infection may impact the gut fungal microbiota in an unknown manner since a shift in the bacterial gut microbiota has been established for infants suffering from RSV infection [52].

In our previous study we showed that after introducing antibiotics the bacterial diversity and richness dropped significantly at 1–4 days, and then it rose again after cessation of antibiotic treatment. Additionally, we observed a rapid change in the bacterial composition during the antibiotic course with bifidobacteria already replaced by enterobacteria 4 days after the start of antibiotics. After the treatment ended, the bacterial microbiota started to recover by an increase of bifidobacteria, but there was still an increase of Firmicutes 6 months after the start of antibiotics [19]. Here we show that the fungal diversity and richness is higher in the antibiotic-treated group at both 3–5 days, 1–2 weeks and still at the last timepoint, >6 weeks after start of antibiotics compared to the controls, with an even more significant difference between macrolide and control patients. These results agree with a recent adult study, where it was observed that a single course of antibiotics caused an increase in fungal diversity [30]. In other words, as the bacterial diversity decreased, the fungal diversity increased. Previously, it has been observed that the aberrant composition of microbiota caused by antibiotics lasts longer in the gut mycobiota compared to the bacterial microbiota in adults [30]. We observed persisting changes in the fungal composition as well as in the bacterial composition, as shown previously [19].

The gut mycobiota was comparable between the control group and antibiotic-treated group before start of antibiotic treatment, while the antibiotic-treated group already had a higher relative abundance of the bacteria belonging to *Bacteroidaceae*, *Bifidobacterium*, *Enterococcus*, *Bacillus* and *Haemophilus* at baseline in our previous analysis with the same cohort [19]. This could be attributed to the overall lower abundance of the fungal versus the bacterial communities. After the start of antibiotic treatment, the relative abundance of *Candida* was observed to be significantly more abundant in the antibiotic-treated infants compared to controls, while *Saccharomyces* was more abundant in the controls. The same trend was seen throughout. We also found that the fungal phylum Basidiomycota was relatively more abundant in the antibiotic-treated group at the end of the study. Further, we found that Basidiomycota was more abundant in the macrolide-treated infants compared to controls at >6 weeks after start of treatment, while *Malassezia* was more abundant in both amoxicillin- and macrolide-treated infants. Basidiomycota has been observed to be more abundant in IBD previously [16], suggesting a persisting fungal dysbiosis more than 6 weeks after antibiotics in these patients. In the bacterial gut microbiota, we found macrolides to be more detrimental than amoxicillin, and together with the fungal gut microbiota, these and earlier results indicate that macrolides might have a more persistent effect on the gut microbiota compared to amoxicillin [25].

While being a part of the healthy gut microbiota [13], *Candida* has been observed to be elevated in inflammatory diseases [2,3,4,5,6,7,8], and further increased in antibiotic-treated mice [28,29]. *Candida* is a yeast characterized by polymorphism, including both different growth forms and morphologies, of which particularly hyphae formation is observed to be associated with virulence [53]. In a recent study it was found that bacterial metabolites control the growth and virulence of *Candida albicans* by limiting hyphae formation [30], which also has been observed previously [54]. Since antibiotics do not directly affect fungi, the observed changes in the gut mycobiota are likely mediated by the changing bacterial composition. Our results strongly suggest that commensal gut bacteria regulate the fungi, keeping them in check. When the bacteria are disrupted by antibiotics, fungi, especially *Candida*, have the opportunity to grow. Unfortunately, we do not have data on the absolute abundances of the fungi, but other studies have shown that overgrowth of *Candida* is a common side-effect of antibiotic treatments [30]. Further, *Candida* is elevated after antibiotic treatment and prevents re-establishment of *Lactobacillus* spp. in microbiome-disturbed mice [55]. It has been studied that gut fungi, particularly *C. albicans*, induce Th17 cells [56,57]. Th17 regulation is important in the development of chronic inflammatory diseases, such as IBD [58]. However, it is debated whether *Candida* is a driver of inflammation, or whether it simply benefits from inflammation [12].

All infants included in this study had an RSV infection at baseline and therefore the observed differences between the study groups are unlikely to be due to the infection, but due to the antibiotics. Additionally, the infants had not received any antibiotics prior to being included in this study, i.e., all infants were antibiotic-naïve at recruitment. The relatively low number of patients, particularly the controls, is a limitation of this study, as in most pediatric studies. Finally, the fungal data were relative abundances and highly zero-inflated, meaning there was a high number of samples devoid of the given fungal taxa, which was considered by taking prevalence into account in all statistical analyses. Zero-inflation is observed in most, if not all, fungal microbiota studies, and therefore statistical tests designed for normally distributed data may produce unreliable results.

## 5. Conclusions

Introduction of only a single course of antibiotics during the development of the gut mycobiota in infants caused a shift in the fungal gut composition, characterized by a higher relative abundance of *Candida*, and higher diversity and richness. This could indicate that aberrant gut mycobiota composition after antibiotic treatment could, together with the bacterial microbiota, be a cause of the long-term effects that antibiotics have on human health.

## Figures and Tables

**Figure 1 jof-08-00328-f001:**
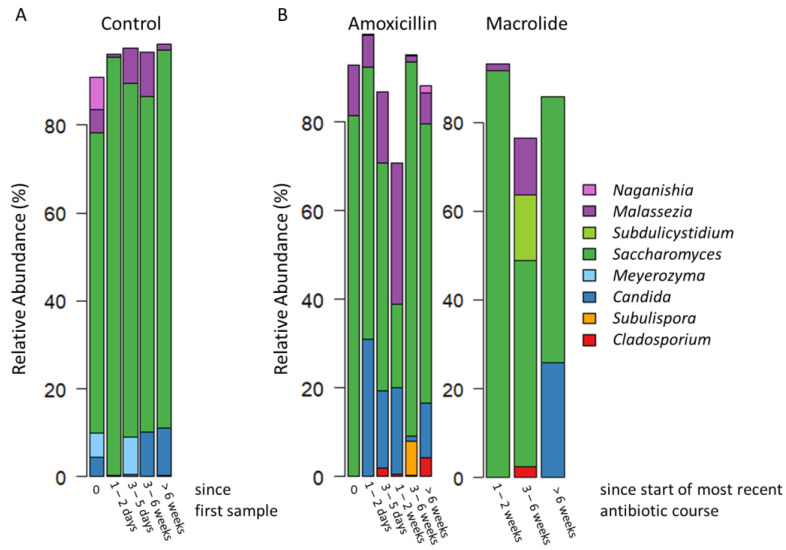
The gut fungal microbiota (mycobiota) composition presented as mean relative abundances of the most abundant fungal genera (presented on the right side of the figure) in (**A**) infants that were antibiotic-naïve throughout the follow-up and (**B**) infants that were antibiotic-naïve at baseline (marked as 0) but received amoxicillin or macrolide antibiotic treatment during follow-up. For the infants who received multiple antibiotics, the timepoints are calculated as time since the most recent antibiotic. All macrolide-treated patients received at least one course of amoxicillin prior to the macrolide course. Detailed results of the statistical differences between the antibiotic-treated and antibiotic-naïve groups are presented in Appendix A.

**Figure 2 jof-08-00328-f002:**
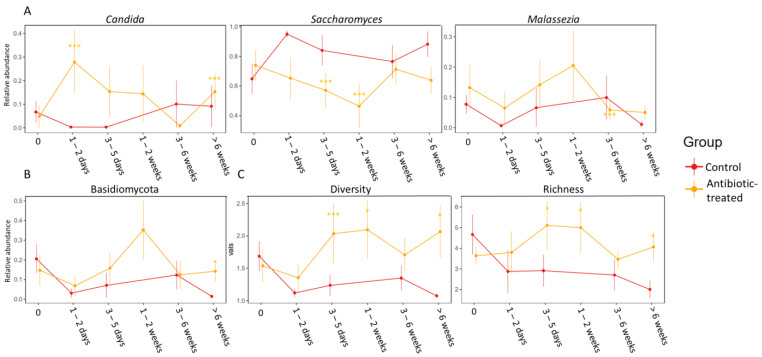
Differences between the control group and antibiotic-treated group over 6 timepoints starting before antibiotic treatment (marked as 0) stratified by (**A**) differences in the relative abundances at genus level, (**B**) at phylum level in the gut fungal microbiota (mycobiota), and further (**C**) differences in diversity and richness between control (red) and antibiotic-treated group (yellow). Significant *p*-values marked as * <0.05 and *** <0.001. All antibiotics are grouped together in the antibiotic group and for the infants who received multiple antibiotics, the timepoints are calculated as time since the most recent antibiotic. Detailed results of the statistical differences between the groups are presented in Appendix A.

**Table 1 jof-08-00328-t001:** Number of infants stratified by control, amoxicillin and macrolide divided into time since most recent course of antibiotics or since first sample. All infants were antibiotic-naïve at timepoint 0.

Time Since Most Recent Course of Antibiotics or Since First Sample	0 Days	1–2 Days	3–5 Days	1–2 Weeks	3–6 Weeks	>6 Weeks
Control (no. of infants—16; no. of samples—50)	14	8	9	0	10	9
Amoxicillin (no. of infants—21; no. of samples—51)	10	9	8	5	9	10
Macrolide (no. of infants—4; no. of samples—12)	0	0	0	3 *	5 *	4

* one cephalosporin.

**Table 2 jof-08-00328-t002:** Patient characteristics stratified by control and antibiotic use.

Patients	All (N = 37)	Antibiotics during Admission (N = 21)	No Antibiotics during Admission (N = 16)
Antibiotic-naïve at recruitment, N(%)	37 (100)	21 (100)	16 (100)
Female, N(%)	15 (40.5)	7 (33)	8 (50)
Age at hospitalization, median (range, months)	2.3 (0.8–9.3)	2.5 (0.8–9.3)	2.1 (0.8–6.8)
Vaginal delivery, N(%)	31 (83.8)	17 (81.0)	14 (87.5)
Exclusive or partial breast-feeding, N(%)	32 (86.5)	17 (81.0)	15 (93.8)
Birth weight, median (range, kg)	3.7 (2.5–4.9)	3.5 (2.5–4.9)	3.8 (2.7–4.5)
Length of hospital stay, median (range, days)	4 (1–9)	4 (2–9)	4 (1–7)
Annotated samples with >100 reads, N	113	63	50
Fecal samples obtained and successfully annotated, median (range, N/patient)	3 (1–6)	3 (1–5)	3.5 (1–6)
Length of follow-up, median (range, months)	3 (0–9.5)	3 (0–9.5)	2 (0–6)

## Data Availability

The ITS rDNA amplicon sequences of this study are available at the ENA database (PRJEB50378).

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
