# Peer review of "The Effect of Antibiotics on the Infant Gut Fungal Microbiota"

_jof, 2022, doi:10.3390/jof8040328_

Round 1

Reviewer 1 Report

Dear Authors,

After the review process, I have several comments: you should clearly present the aim of the paper in the abstract and the last paragraph of the introduction; you should be rewritten because now is a part of the introduction; you should include numerical data in the abstract; you should include statistical data in the Materials and Methods section; it should be the last section here; The discussion section should be expanded because the increased administration of antibiotics determines the prevalence of antibiotic resistance, gut bacterial strains, and sustains dysbiosis, role in the stress response, in developing pathologies, and in inflammatory progression. These data will create a  link between obesity, microbiota dysbiosis, and neurodegenerative pathogenesis; figures and/or tables do not have statistical data, clear mentioned.

Best regards.

Author Response

Response to comments by reviewer 1.

We want to thank the reviewer for the thorough review and insightful comments. Based on your comments we now present an improved and clearer revised version of our manuscript! The point-by-point response to each comment is found below.

Dear Authors,

After the review process, I have several comments:

you should clearly present the aim of the paper in the abstract and the last paragraph of the introduction; you should be rewritten because now is a part of the introduction;

Response: We agree that the aims could have been more clearly stated, and we have therefore changed line 17 of the abstract to “Here, the aim was to investigate the impact…”. Additionally, the aims are now presented as a separate section at the end of the introduction (page 2, lines 67-80) in the revised manuscript.

you should include numerical data in the abstract; you should include statistical data in the Materials and Methods section; it should be the last section here;

Response: Thank you for your comment. We have now added p-values (or range of) to the abstract. However, the number of allowed words is limited to around 200, and therefore we could not elaborate further on the more exact results in the abstract.

We present the statistical analyses used in this study in the section Materials and Methods, marked with subtitle 2.4. Statistical analyses. In the guidelines of Journal of Fungi they state: “Materials and Methods: They should be described with sufficient detail to allow others to replicate and build on published results. New methods and protocols should be described in detail while well-established methods can be briefly described and appropriately cited. Give the name and version of any software used and make clear whether computer code used is available. Include any pre-registration codes.”. Therefore, we are unsure what the referee meant with this comment as we find that we have followed the instructions and added all data and results in the results section.

The discussion section should be expanded because the increased administration of antibiotics determines the prevalence of antibiotic resistance, gut bacterial strains, and sustains dysbiosis, role in the stress response, in developing pathologies, and in inflammatory progression. These data will create a  link between obesity, microbiota dysbiosis, and neurodegenerative pathogenesis;

Response: Thank you for your insight on this. However, the Reviewer 2 asked to expand the discussion further and the Reviewer 3 asked to shorten and simplify the discussion. Therefore, we could not extend the discussion regarding increased administration of antibiotics and antibiotic resistance as this was not the focus of our study. Also, our results do not imply a link between obesity or neurodegenerative pathogenesis. Additionally, these are extremely rare in infants, that were our main focus in this study. We have now revised the discussion, hopefully making it clearer and easier to read.

figures and/or tables do not have statistical data, clear mentioned.

Response: Thank you for this comment. We have now added all significant results from the statistical tests done on the relative abundance data to supplementary tables and referred to these in the figure legends, making particularly figure 1 easier to interpret. However, all statistically significant results are marked with * in Figure 2. We hope these changes will clarify the figures.

Reviewer 2 Report

The article is well designed and provides an answer to the question that antibiotics impact the infant gut fungal microbiota. However, the problem remains that healthy children were not included in the study, but only children with respiratory syncytial virus (RSV) infection. can the authors explain in more detail by citing the literature that RSV has no effect on the microbiota?

Author Response

Response to comments by reviewer 2. The response is outlined in italics below.

The article is well designed and provides an answer to the question that antibiotics impact the infant gut fungal microbiota. However, the problem remains that healthy children were not included in the study, but only children with respiratory syncytial virus (RSV) infection. can the authors explain in more detail by citing the literature that RSV has no effect on the microbiota?

Response: We want to thank the reviewer for the thoughtful review and insightful comment. Based on your comments we now present an improved and clearer revised version of our manuscript!

All infants were suffering from respiratory syncytial virus (RSV) infection, and therefore the differences between the groups are most likely due to antibiotics, and not due to the RSV infection (stated on page 9, lines 320-322). RSV can cause some changes in the gut microbiota as stated in the discussion on page 8, lines 255-259: “It is also worth noting that the infants in this study were not healthy since both antibiotic-treated patients and controls had an active RSV infection at baseline and can therefore not fully be compared to the healthy infants in other studies. Viral infection may impact the gut fungal microbiota in an unknown manner since a shift in the bacterial gut microbiota has been established for infants suffering from RSV infection [53].“. To our knowledge, there are no studies on the effect of RSV on the gut mycobiota, but since the bacterial microbiota is affected, the fungi also are highly likely affected. The important aspect here, however, is that both the antibiotic-treated infants, and those who remained antibiotic-naïve throughout the study all suffered from RSV infection. Therefore, if the RSV would cause changes in the gut microbiota, the changes would be present in both groups, and therefore the results between the groups will, with high certainty be due to the antibiotics. If we would compare RSV patients to healthy controls the study aims would be different as the infection itself would be the focus, not the antibiotics. To conclude, in this cohort we can compare antibiotic-treated to antibiotic-naïve infants who all have RSV. What we cannot do, and have not done, is to compare RSV patients to healthy controls.

We are sorry if this remained unclear, and we have done some changes in the wording of the manuscript to clarify. We have however not further elaborated on the discussion, due to reviewer 3 asking to shorten the discussion and also stating that the possible impact of RSV is well described in the discussion.

Reviewer 3 Report

The Authors described the impact of antibiotic treatment on gut mycobiota of infant. They enrolled 37 patients, 16 of them were not administered antibiotics and, as a consequence, considered as control group.

The findings of this study appear very interesting and presented in an adequate manner. Moreover, the Authors well described the limit of their study (ie: the possible impact of RSV infection on the "basal mycobiota").

I suggest:

  • please, use the same term in the whole paper ("mycobiota" or "fungal microbiota")
  • please simplify and shorten the discussion; it seem a bit redundant in some part and difficult to read.

Author Response

Response to comments by reviewer 3. The response is outlined in italics below.

The Authors described the impact of antibiotic treatment on gut mycobiota of infant. They enrolled 37 patients, 16 of them were not administered antibiotics and, as a consequence, considered as control group.

The findings of this study appear very interesting and presented in an adequate manner. Moreover, the Authors well described the limit of their study (ie: the possible impact of RSV infection on the "basal mycobiota").

I suggest:

  • please, use the same term in the whole paper ("mycobiota" or "fungal microbiota")
  • please simplify and shorten the discussion; it seem a bit redundant in some part and difficult to read.

Response: We would like to thank the reviewer for the thorough revision of our manuscript, and for the comments. We are happy that the reviewer found our manuscript interesting. Based on your comments we are certain that we have an improved and clearer revised version of our manuscript!

We agree that the same term for fungal microbiota should be used, and since “mycobiota” is an established term, we have now used that throughout the study with an exception in the cases we are referring to both bacterial and fungal gut microbiota, where we found that both should be specified for clarity. However, we still kept the “fungal gut microbiota” in the title of the manuscript, since it provides more information.

We have now revised the discussion further and hope the changes will make it easier to read. The changes are seen in the revised manuscript. However, the Reviewer 2 asked to expand the discussion further and the Reviewer 3 asked to shorten and simplify the discussion. Therefore, we have decided not to remove any parts, but to overall revise and clarify.

Reviewer 4 Report

The paper submitted by Ventin-Holmberg is focused on the analysis of changes in mycobiota composition in children after the treatment which chosen antibiotics. In general the paper is well written and the subject is interesting. In my opinion it should be published in Journal of Fungi after addressing some minor comments:

1) the quality of Fig 2 should be improved - in its present form the figure is difficult to read (graphs in particular)

2) what about showing statistically significant differences between analysed groups in Fig 1? what are the p values?

Author Response

Response to comments by reviewer 4. The point-by-point response is found in italics below.

The paper submitted by Ventin-Holmberg is focused on the analysis of changes in mycobiota composition in children after the treatment which chosen antibiotics. In general the paper is well written and the subject is interesting. In my opinion it should be published in Journal of Fungi after addressing some minor comments:

Response: We want to thank the reviewer for the revision. Based on your comments we now present an improved and clearer revised version of our manuscript!

1) the quality of Fig 2 should be improved - in its present form the figure is difficult to read (graphs in particular)

Response: As the manuscript itself does not include the pdf:s, the quality of the figures are unfortunately too low. We have also added the pdf:s of the figures and there the quality is significantly better, and these are included in the final version when published. We have, however, also revised Figure 1, making the graphs and legends bigger. We are certain that the changes made, together with the improved quality of the pdf version, will make the figure easier to read, and hopefully the reviewers also will get access to the pdf versions with better quality.

 2) what about showing statistically significant differences between analysed groups in Fig 1? what are the p values?

Response: We have now added all the statistically significant results to the supplement and referred to this in Fig 1. The purpose of Fig 1 is to present the gut mycobiota composition, while the differences between the groups are presented more clearly in Fig 2. However, we hope adding the supplementary table to the revised manuscript helps in the interpretation as well.

Round 2

Reviewer 1 Report

No other comments compared to the first review.

No improvement regarding dysbiosis after antibiotic treatment and other health effects.

Author Response

Response to reviewer 1, second revision.

No other comments compared to the first review.

No improvement regarding dysbiosis after antibiotic treatment and other health effects.

Response: We thank the reviewer for pointing out that the impact of dysbiosis to long-term health problems could be mentioned more clearly. While we still are slightly unsure what the exact request of the reviewer is, we now have revised the introduction elaborating on the dysbiosis created by antibiotics by adding the following sentence: “The use of antibiotics during the critical time from conception to 2 years of age has been reported to cause both short-term problems including dramatic shifts of the gut fungal and bacterial microbiota and even sepsis, and further long-term consequences such as  asthma, atopy, pediatric onset of Crohn’s disease and metabolic diseases.” and also cited a recent review on this topic (number 21 in revised version: Aires J. First 1000 Days of Life: Consequences of Antibiotics on Gut Microbiota. Front Microbiol. 2021 May 19;12:681427. doi: 10.3389/fmicb.2021.681427.). The changes are seen on page 2 in the Introduction. However, as the other reviewers commented that Discussion was lengthy, we did not add any more discussion. We are certain that the revised introduction provides a clearer background to the study.